# Substrate Selectivity of a Novel Amylo-α-1,6-glucosidase from *Thermococcus gammatolerans* STB12

**DOI:** 10.3390/foods11101442

**Published:** 2022-05-16

**Authors:** Yamei Wang, Yixiong Tian, Xiaofeng Ban, Caiming Li, Yan Hong, Li Cheng, Zhengbiao Gu, Zhaofeng Li

**Affiliations:** 1School of Food Science and Technology, Jiangnan University, Wuxi 214122, China; jlauwym@163.com (Y.W.); 7180112083@stu.jiangnan.edu.cn (Y.T.); banxiaofeng521@163.com (X.B.); caimingli@jiangnan.edu.cn (C.L.); hongyan@jiangnan.edu.cn (Y.H.); chenglichocolate@163.com (L.C.); zhengbiaogu@jiangnan.edu.cn (Z.G.); 2Key Laboratory of Synergetic and Biological Colloids, Ministry of Education, Wuxi 214122, China; 3Collaborative Innovation Center of Food Safety and Quality Control, Jiangnan University, Wuxi 214122, China

**Keywords:** amylo-α-1,6-glucosidase, substrate selectivity, pullulanase, isoamylase, debranching efficiency

## Abstract

Amylo-α-1,6-glucosidase (EC 3.2.1.33, AMY) exhibits hydrolytic activity towards α-1,6-glycosidic bonds of branched substrates. The debranching products of maltodextrin, waxy corn starch and cassava starch treated with AMY, pullulanase (EC 3.2.1.41, PUL) and isoamylase (EC 3.2.1.68, ISO), were investigated and their differences in substrate selectivity and debranching efficiency were compared. AMY had a preference for the branched structure with medium-length chains, and the optimal debranching length was DP 13–24. Its optimum debranching length was shorter than ISO (DP 25–36). In addition, the debranching rate of maltodextrin treated by AMY for 6 h was 80%, which was 20% higher than that of ISO. AMY could decompose most of the polymerized amylopectin in maltodextrin into short amylose and oligosaccharides, while it could only decompose the polymerized amylopectin in starch into branched glucan chains and long amylose. Furthermore, the successive use of AMY and β-amylase increased the hydrolysis rate of maltodextrin from 68% to 86%. Therefore, AMY with high substrate selectivity and a high catalytic capacity could be used synergistically with other enzyme preparations to improve substrate utilization and reduce reaction time. Importantly, the development of a novel AMY provides an effective choice to meet different production requirements.

## 1. Introduction

Starch, the second-largest biomass resource on earth, is an essential raw material for food manufacturing [1], paper [2], textile [3], and biofuel [4] industries. Starch consists of two polymers, amylose and amylopectin [5]. Amylose is a polysaccharide chain of D-glucosyl linked via α-1,4-glycosidic bonds [6], whereas amylopectin is a highly branched polysaccharide with α-1,4-glucosidic linkages in the glucan chain and α-1,6-glucosidic linkages at the branch points after every 20 to 30 glucose units [7].

At present, most amylases that hydrolyze α-1,4-glycosidic bonds cannot or can only slowly cleave α-1,6-glycosidic bonds. Therefore, starch liquefaction products contain a large amount of amylopectin and limit dextrin, resulting in a low starch utilization rate and high amylase addition. However, debranching enzymes can efficiently hydrolyze the α-1,6-glycosidic bonds in amylopectin and limit dextrin, thereby significantly improving starch utilization and reducing the number of other enzyme preparations used [8]. Therefore, the utilization of debranching enzymes that efficiently hydrolyze α-1,6-glycosidic bonds is indispensable to enhancing the efficiency of amylopectin hydrolysis and has been widely used in the industrial production, of glucose syrup [9], maltose syrup [10], cyclodextrin [11,12] and resistant starch [13,14].

Debranching enzymes usually include pullulanase (EC 3.2.1.41, PUL), isoamylase (EC 3.2.1.68, ISO) and amylo-α-1,6-glucosidase (EC 3.2.1.33, AMY) [15]. In recent years, a great deal of research has focused on PUL and ISO, which have different substrate specificities [16,17]. PUL can hydrolyze the α-1,6-D-glycosidic bonds in pullulan, β-limit dextrin, and branched oligosaccharides, generating maltose, maltotriose and linear oligosaccharides [17,18], whereas ISO cannot hydrolyze α-1,6-glycosidic bonds with only two glucose groups. The smallest unit of the substrate side chain contains at least three or four glucose residues [19]. These differences indicated that PUL selectively cleaved branched chains with short-chain glucose residues, whereas ISO preferred branched chains with long chain glucose residues. However, starch was hydrolyzed by α-amylase to produce branched dextrin of small molecular weight, and medium length branched chains during the starch liquefaction process. PUL and ISO could not effectively improve the raw material utilization and production efficiency since the products of starch liquefaction were not their optimal substrates. Therefore, this study aims to develop a novel debranching enzyme that could specifically hydrolyze medium-length branched chains. By cloning and expressing AMY in *Escherichia coli* and studying its substrate selectivity, it is expected to lay a solid foundation for the industrial application of the enzyme.

AMY has different substrate specificities from PUL and ISO, and little has been reported in the literature. Previous studies mainly sought to explore the expression and physicochemical properties of this enzyme. AMY from *mammalian tissue* and *yeast* have been reported to have two distinct activities, maltooligosaccharide transferase and amylo-1,6-glucosidase [20]. Furthermore, *bacterial* AMY has been documented to specifically hydrolyze α-1,6-glycosidic bonds to release maltotetraose and maltodextrin [21,22,23]. Interestingly, Dauvillee et al. cloned and expressed the gene encoding *GlgX* and found that the enzyme could specifically hydrolyze the side chains consisting of three or four glucose residues [21]. Furthermore, Park et al. cloned and expressed the *TreX* gene derived from *Sulfolobus Solfataricus* P2 in *E*. *coli*, which showed strong substrate specificity and high selectivity for side chains consisting of six or more glucose residues [24]. Although some genes encoding AMY from different sources have been cloned and expressed, the properties of the related enzymes have not been elucidated. In particular, their selectivity on substrate species and debranching length are unclear. We first cloned the *amy* gene (NCBI number: NC_012804.1) from the archaea *Thermococcus gammatolerans* STB12 and successfully expressed the *amy* gene in *E. coli.* The optimum temperature of AMY was 70 °C, and the optimum pH was 4.0, which met the requirements of heat resistance and acid resistance of debranching enzymes in a saccharification reaction.

In this study, we analyzed the activity of the novel AMY against different substrates to characterize their substrate specificity. We also compared the differences in hydrolytic products of AMY, PUL, and ISO, illustrating the selectivity and debranching efficiency of the debranching enzymes for different branched structures. Importantly, our results could guide the selection of specific debranching enzymes for various industrial processes.

## 2. Materials and Methods

### 2.1. Materials

*E. coli* JM109 and *E. coli* BL21(DE3) strains were used for gene cloning and expression, respectively. The recombinant plasmid pET-20b(+) bearing the *amy* gene from *Thermococcus gammatolerans* STB12 was constructed in this study. Maltodextrin (*M*_W_*:* 1.0 × 10^4^–10.0 × 10^4^ g/mol) was obtained from Roquette Frers (Lestrem, France), Pullulan (*M*_W_*:* 1.0 × 10^3^–10.0 × 10^3^ g/mol) was obtained from Bailingwei Co., Ltd. (Shanghai, China), waxy corn starch (*M*_W_*:* 1.0 × 10^6^–10.0 × 10^6^ g/mol) was obtained from Cargill Asia Pacific Food Systems Co., Ltd. (Beijing, China), cassava starch (*M*_W_: 1.0 × 10^6^–10.0 × 10^6^ g/mol) was obtained from Hongfeng Starch Co., Ltd. (Guangxi, China), and corn starch (*M*_W_: 1.0 × 10^6^–10.0 × 10^6^ g/mol) was obtained from Hebei Yufeng Industry Group Co., Ltd. (Hebei, China). Glycogen (*M*_W_: 1.0 × 10^6^–10.0 × 10^6^ g/mol) and dimethyl sulfoxide (DMSO) were purchased from Sinopharm Group Co., Ltd. (Shanghai, China). *Klebsiella pneumoniae* pullulanase (EC 3.2.1.41, 3000 U/mL) and sodium acetate were purchased from Aladdin Reagent Co., Ltd. (Shanghai, China). Maize amylopectin (*M*_W_: 1.0 × 10^7^–10.0 × 10^7^ g/mol) (Cas: 9037-23-4), potato amylopectin (*M*_W_: 1.0 × 10^7^–10.0 × 10^7^ g/mol) (Cas: 9037-22-3), *Pseudomonas amyloderamosa* isoamylase (EC 3.2.1.68, 10,000,000 U/mL), and sodium hydroxide solution were purchased from Sigma-Aldrich Co. Ltd. (St. Louis, MO, USA). The barley β-amylase (EC 3.2.1.2, 700,000 U/mL) was purchased from Yuanye Biotechnology Co., Ltd. (Shanghai, China). All reagents were of analytical grade unless otherwise stated.

### 2.2. Expression and Purification of AMY

AMY was produced in the *E. coli* BL21(DE3)-harboring plasmid *amy*/pET-20b(+). The crude enzyme was filtered by a 0.45 μm filter, and the His_6_-tagged AMY was purified with HisTrap HP. The purification process was carried out as follows: (1) Equilibrate the nickel column with buffer A (10 mM Tris-HCl, 500 mM NaCl, pH 7.5) at a flow rate of 2.0 mL/min. After loading, continue to equilibrate with buffer A. (2) Elute the nickel column with buffer B (10 mM Tris-HCl, 500 mM NaCl, 500 mM imidazole, pH 7.5) at a flow rate of 1.5 mL/min. The eluent was collected and verified using SDS-PAGE. The concentration of AMY was determined using a Bradford kit purchased from Generay (Shanghai, China), taking bovine serum albumin (BSA) as standard.

### 2.3. Enzyme Activity Assays

The AMY activity was determined according to the method described by Yoshinori et al. [25], with a slight modification. The mixture containing 700 μL of 10 mg/mL DE6 (DE, dextrose equivalent) maltodextrin and 150 μL of sodium acetate buffer (500 mM, pH 4.0) was incubated at 70 °C for 10 min. Then, 150 μL AMY was added to the substrate solution and mixed thoroughly. The reaction was conducted at 70 °C for 15 min and terminated by boiling for 20 min. A color change was observed when 100 μL of the resulting solution was mixed with an equal volume of 0.01 M iodine-potassium iodide solution, and diluted with deionized water to 5.0 mL. The solution was allowed to stand for 15 min at room temperature (25 °C) and then its absorbance was measured at 610 nm. One unit of enzyme activity was defined as the amount causing an 0.1 increase in 1 h at A_610_ nm using DE6 maltodextrin. The sample without any enzyme was set as a control.
$$\mathrm{Units}/\mathrm{mL}=\frac{\left(A_{610\;nm}Test-A_{610\;nm}Blank\right)\left(4\right)\left(5\right)}{\left(0.1\right)\left(0.015\right)}$$
where 5 = Total volume (in milliliters) of assay; 4 = Time (in minutes) conversion factor from 15 min to 60 min; 0.1 = Increase in A_610_ nm per hour; 0.015 = Volume (in milliliter) of enzyme used.

### 2.4. Analysis of Substrate Specificity

The substrate specificity was determined using glycogen, pullulan, rice starch, cassava starch, wheat starch, potato amylopectin, maize amylopectin, waxy corn starch, corn starch, DE2 maltodextrin, DE4 maltodextrin, and DE6 maltodextrin. The 3,5-dinitrosalicylic acid (DNS) assay for reducing sugar was used to quantify the hydrolytic activity against different substrates [26,27]. Assay mixtures contained 150 μL AMY and 850 μL 10 mg/mL dextrin or starch solution, while the dextrin or starch solution was prepared with 100 mM sodium acetate buffer (pH 4.0). The mixtures were incubated at 70 °C for 15 min, and the reaction was terminated by adding 1 mL of DNS. The resulting solution was boiled in a boiling water bath for 5 min and immediately cooled in ice water. Finally, the absorbance of this mixture was measured at 540 nm. One unit (U) of enzyme activity was defined as the amount required to produce 0.01 μmol of reducing sugar (in terms of glucose) per minute. The highest activity assayed using the optimum substrate was set at 100%. The correlation between glucose concentration and OD value can be expressed as follows:Regression equation: y = 0.434x + 0.9037
Correlation coefficient: R^2^ = 0.9939

### 2.5. Preparation of the Debranched Dextrin and Starch

DE6 maltodextrin, waxy corn starch, and cassava starch (1 g, dry basis) were dissolved in deionized water (100 mL), gelatinized in boiling water at 100 °C for about 20 min, and cooled at room temperature. The mixture was equilibrated at 40 °C, 45 °C, and 70 °C, in the water bath for 10 min, then AMY (25 U/g, 70 °C, pH 4.0), PUL (2 U/g, 45 °C, pH 4.5) and ISO (10 U/g, 40 °C, pH 3.5) were added. Hydrolysis was carried out at different temperatures for 24 h with constant shaking at 160 rpm. After incubation for 0, 6, and 24 h, the reaction was terminated in a boiling water bath for 20 min. The supernatant was harvested by centrifugation at 5000× *g* (10,000 rpm) for 20 min; the final debranched samples were collected by freeze-drying. The samples prepared without enzymes were set as a control.

### 2.6. Chain Length Distribution Analysis

High-performance anion-exchange chromatography coupled with pulsed amperometric detection (HPAEC-PAD) (Dionex ICS-5000, Thermo Scientific, Waltham, MA, USA) was used to analyze the chain length distribution of the debranched products [28]. The 10 mg samples were dissolved in 2 mL ultra-pure water and heated in a boiling water bath for 20 min. Subsequently, the liquid mixtures were centrifuged at 5000× *g* (10,000 rpm) for 1 min, and the debranched samples passed through a 0.22 μm membrane filter. The filtrate was injected into the HPAEC-PAD system equipped with a Carbopac^TM^ PA200 (3 × 250 mm) column maintained at 35 °C using a flow rate of 0.5 mL/min. The column was equilibrated in 250 mM sodium hydroxide, 1 M sodium acetate, and ultra-pure water.

### 2.7. Molecular Weight Distribution Analysis

Gel permeation chromatography (GPC) was used to determine the molecular weight distribution (MWD) of the substrates treated with different debranching enzymes. MWD was measured according to the method reported by Liu et al. [29] with some modifications. The phenogel columns used were Styragel HR3 (M_W_: 500–30,000), Styragel HR4 (M_W_: 5000–600,000), and Styragel HMW7 (M_W_: 500,000–1 × 10^8^) (Waters, Inc., Torrance, CA, USA), the flow rate of the eluent was 0.5 mL/min, the eluent comprised 99.5 wt% DMSO and 0.5 wt% LiBr, and the column temperature was maintained at 50 °C. The 10 mg debranched samples were dissolved in 2 mL of the eluent and heated in a boiling water bath for 12 h by continuous stirring. The debranched samples were passed through a 0.22 μm nylon filter, and the filtrate was injected into a Shimadzu HPLC/GPC instrument (CTO-20A; Shimadzu Corporation, Kyoto, Japan) equipped with a RI detector (Wyatt Technologies, Santa Barbara, CA, USA).

### 2.8. β-Amylolysis Limit Analysis

The degree of β-amylolysis was determined according to the method described by Kong et al. [30]. Each debranched sample (10 mg) was dissolved in 2 mL 50 mM acetate buffer (pH 5.0) by heating in a boiling water bath for 30 min, and stirring continuously. After incubation at 50 °C for 10 min, the products were further hydrolyzed at 50 °C for 24 h by adding β-amylase (50 U/mg substrates). All reactions were terminated by boiling for 20 min. The maltose content in each sample was determined using HPAEC-PAD [31]. The β-amylolysis limit was calculated according to the formulas established by Shen et al. [32]. The degree of β-amylolysis was calculated by the ratio of the total mass of maltose produced over to the total mass of total carbohydrates (dry basis).

### 2.9. Statistical Analysis

The result was expressed as mean ± standard deviation (SD) from triplicate experiments. The statistical analysis was conducted with SPSS statistical software version 25.0 (SPSS Inc., Chicago, IL, USA). The comparison of treatment means was determined using Duncan’s test at a 5% level of significance.

## 3. Results and Discussion

### 3.1. Expression and Purification of AMY

The SDS-PAGE of AMY was purified using the HisTrap HP affinity columns as shown in Figure 1. The protein band is single and clear, achieving electrophoresis purity. The molecular weight (about 66 kDa) is close to the theoretical molecular weight. The enzyme activity and protein concentration of AMY were measured, and the calculation results are shown in Table 1. The specific activity of AMY purified by HisTrap was 733.3 U/mg, and the recovery rate was 37.2%.

### 3.2. Analysis of Substrate Specificity

The substrate specificity of AMY was analyzed using different substrates such as maltodextrin, starch, pullulan, and glycogen (Figure 2). The optimal AMY activity (360 U/mg) towards DE6 maltodextrin was obtained at pH 4.0 and 70 °C. The enzyme activity assayed using DE6 maltodextrin was defined as 100% activity, and the relative activities using DE4 maltodextrin and DE2 maltodextrin were 75% and 61%, respectively. However, the relative activity of AMY was below 50% when starch was used as the substrate. The results illustrated that AMY had a low affinity on large molecular weight substrates; the relative activity of AMY to glycogen was 57%, which was less than the relative activity for maltodextrin due to the fine structure of glycogen with high branching density and short branched chains [33,34]. In a nutshell, AMY showed high substrate selectivity; it preferentially cleaved maltodextrin with moderate molecular weight and medium-length branched chains, while it exhibited low hydrolysis activity on starch and pullulan.

### 3.3. Chain Length Distribution

The chain length distribution of the DE6 maltodextrin debranched by AMY, PUL and ISO was analyzed using HPAEC-PAD. The profiles are shown in Figure 3. According to the degree polymerization (DP), the amylopectin side chain can be divided into four fractions: A chains (DP 6–12), B1 chains (DP 13–24), B2 chains (DP 25–36), and B3 chains (DP ≥ 37) [35,36]. The proportions of each fraction in debranched samples are listed in Table 2. Compared to the control, the chain length distribution and average chain length (CL) significantly changed after AMY treatment. During the early stages, the proportions of DP < 6 and DP 6–12 fractions significantly decreased from 38.57% to 31.74% and 43.20% to 42.20%, respectively. In addition, the proportion of DP 13–24 fractions significantly increased from 17.09% to 23.12%. As the reaction continued, the proportion of DP 13–24 fractions decreased, while DP < 6 fractions increased significantly. In contrast, AMY had a limited hydrolytic effect on waxy corn starch and cassava starch, and the chain length distribution could not be detected by HPAEC-PAD (data not shown). This finding was due to the high branching density and long branched chains of waxy corn starch and cassava starch [37], thus resulting in steric hindrance between enzyme and substrate. The results suggested that the AMY could selectively hydrolyze α-1,6-glycosidic bonds, and showed vigorous hydrolysis activity on maltodextrin. Furthermore, AMY selectively cleaved side chains with DP 13–24.

Compared with AMY, PUL and ISO had different debranching specificities. The chain length distribution and average chain length changed significantly after PUL treatment, which resulted in an altered maximum peak value. Interestingly, the glucose content increased dramatically after PUL treatment, indicating that PUL hydrolyzed not only α-1,6-glycosidic bonds but also α-1,4-glycosidic bonds [38]. The proportion of DP 13–24, DP 25–36, and DP ≥ 37 fractions increased significantly with increasing PUL treatment time. A reasonable explanation may be that PUL randomly cleaved α-1,6-glycosidic bonds at the branching points and α-1,4-glycosidic bonds in the linear chain, generating a mixture of linear chains with different degrees of polymerization. On the other hand, the debranching products of ISO contained a greater proportion of long linear chains than AMY. The proportion of DP 25–36 fractions increased significantly from 1.13% to 10.27%, while the glucose content did not change significantly. The results indicated that ISO tended to cleave α-1,6-glycosidic bonds of the long branched chain in amylopectin, and has no hydrolytic activity on α-1,4-glycosidic bonds. The optimum debranched chain length was DP 25–36.

### 3.4. Molecular Weight Distribution

The MWD of dextrin and starch debranched by AMY, PUL and ISO were determined using GPC. The molecular weight curves are shown in Figure 4. Two peaks were observed in the debranched samples; peak 1 represents the unit chains released by debranching, and peak 2 corresponds to the branched molecules [39]. DE6 maltodextrin was an intermediate product of starch hydrolysis, which contained a certain amount of amylose. The elution behavior of untreated DE6 maltodextrin showed a typical bimodal molecular weight distribution. Waxy corn starch contained almost 100% amylopectin [40], and the elution behavior of amylopectin showed a unimodal distribution, which was different from the typical bimodal molecular weight distribution of normal corn starch containing a high proportion of amylose [41]. Cassava starch has about 80–85% of amylopectin, and a typical bimodal molecular weight distribution was observed due to the presence of a certain proportion of amylose [42].

The MWD of hydrolysates is listed in Table 3. AMY, PUL, and ISO exhibited capacity activity on DE6 maltodextrin, waxy corn starch, and cassava starch; however, all debranching products demonstrated varying degrees of degradation compared to the control. As shown in Figure 4a–c, debranching products of DE6 maltodextrin cleaved by AMY had a relatively narrow molecular weight range and a smaller molecular weight corresponding to the amylopectin peak (peak 2). In addition, the amylopectin was hydrolyzed to produce a large number of linear short amylose and oligosaccharides. On the contrary, AMY debranched waxy corn starch and cassava starch which had a relatively wide molecular weight range and a larger molecular weight corresponding to the amylopectin peak. The molecular weight did not decrease significantly; however, it shifted to the left. The reason was that the amylopectin in waxy corn starch and cassava starch had a fine structure consisting of high branching density and long side chains which resulted in partial hydrolysis of amylopectin to produce branched dextran chains and long amylose [43].

Compared with AMY, the molecular weight of the products debranched by PUL shifted to a lower molecular weight in Figure 4d–f, while the molecular weight of the amylopectin peak (peak 2) decreased (Table 3). After DE6 maltodextrin was treated with PUL for 24 h, its amylopectin peak disappeared, and the polymerized amylopectin was completely decomposed into amylose. In addition, the molecular weight of waxy corn starch hydrolyzed by PUL decreased sharply and switched from a unimodal to a bimodal distribution. The debranched cassava starch also showed the same tendency. The amylopectin was degraded to linear chains, which could be attributed to the fact that PUL not only cleaved a-1,6-glycosidic bonds at the branch points but also randomly hydrolyzed a-1,4-glycosidic bonds, resulting in a significant reduction in molecular weight.

On the contrary, ISO exhibited different substrate selectivity compared with AMY. As shown in Figure 4g–i, ISO had higher hydrolytic efficiency on starch than maltodextrin. The molecular weights of debranched waxy corn starch and cassava starch were significantly decreased. The amylopectin in the waxy corn starch was completely degraded after debranching for 24 h by ISO, whereas ISO showed significantly lower hydrolytic activity on DE6 maltodextrin than AMY. These results indicated that ISO had a high affinity for amylopectin molecules and selectively hydrolyzed the high molecular polymer. Our results were consistent with the findings reported in the literature [19,44].

### 3.5. Analysis of Debranching Efficiency

According to the MWD curves of the debranching products, the ratio of peak 1’s area to the total area represented the relative content of the linear chains. By calculating the percentage of the area under the curve of peak 1 (amylose) to the whole distribution curve (amylose and amylopectin) [40], the debranching efficiencies of AMY, PUL and ISO were obtained. As shown in Figure 5, AMY could hydrolyze most of the amylopectin in DE6 maltodextrin in a short time, and the debranching rate reached 80% at 6 h. As the reaction continued, the hydrolysis rate of AMY increased slowly, and the debranching rate was 88% at 24 h. AMY could not completely hydrolyze the polymerized amylopectin in DE6 maltodextrin because its branched chain included a portion of long or short side chains in addition to medium-length side chains [45]. Furthermore, AMY could slightly hydrolyze waxy corn starch and cassava starch, and the debranching rates were only 24% and 30% for 24 h, respectively. These results showed that AMY tended to hydrolyze the branched chains with medium length (DP 13–24), and a higher proportion of branched chains with medium length would undergo a higher degree of hydrolysis. The rapid hydrolysis of DE6 maltodextrin, waxy corn starch and cassava starch by PUL could be mostly attributed to PUL removing linear glucans from the non-reducing ends of chains by hydrolyzing α-1,4-glycosidic bonds. The presence of numerous short-branched chains promoted the specific binding of PUL with short-branched substrates. Remarkably, the ISO debranching efficiency on DE6 maltodextrin was significantly lower than AMY during the same reaction time. The debranching rate at 6 h and 24 h was 60% and 67%, respectively, which was 20% lower than that of AMY. However, ISO exhibited a strong hydrolytic ability to waxy corn starch and cassava starch, with debranching rates of 100% and 85% for 24 h, respectively. In the process of starch hydrolysis, a large amount of linear amylose and a small number of branched dextran chains exist in the ISO hydrolysate, while a large amount of polymerized amylopectin and a small amount of linear amylose exist in the AMY hydrolysate. It is speculated that ISO randomly hydrolyzes α-1,6-glycosidic bonds at branch points from the outside and inside of amylopectin [13], while AMY slowly hydrolyzes amylopectin from the outside and has little effect on the inside branch points.

### 3.6. Analysis of β-Amylolysis Limit

β-amylase is an exo-amylase, which has been reported to remove maltose units from the non-reducing ends of starch molecules through hydrolysis of α-1,4-glycosidic bonds but could not hydrolyze or stride α-1,6-glycosidic bonds [46]. Therefore, the hydrolysis rate of β-amylase could reflect the debranching degree of substrate molecules, thus mirroring the debranching activity of AMY, PUL, and ISO. As shown in Figure 6, the β-amylase hydrolysis rate of the DE6 maltodextrin was 68%, and the β-amylase hydrolysis rate of the DE6 maltodextrin debranched by AMY and PUL increased to 86% and 94%, respectively. However, no significant change in β-amylase hydrolysis rate was observed when DE6 maltodextrin was debranched by ISO. The β-amylase hydrolysis rates of the waxy corn and cassava starch were 63% and 61% lower than DE6 maltodextrin (68%), respectively. There was no significant change observed for waxy corn and cassava starch treated by AMY; however, the β-amylolysis rate of waxy corn starch and cassava starch debranched by ISO increased to 92% and 86%, respectively. These results were also in agreement with the findings of the debranching efficiency analysis. It proved that AMY had strong substrate selectivity, and specifically hydrolyzed α-1,6-glycosidic bonds in maltodextrin with medium molecular weight, while it exerted no hydrolytic ability on α-1,4-glycosidic bonds. In theory, PUL could not penetrate the interior of the amylopectin molecule due to the dense branching structure, and it was inclined to hydrolyze low molecular weight substrates such as limit dextrin and pullulan. However, PUL could hydrolyze α-1,4-glycosidic bonds randomly and produce a large number of short branched chains, which substantially improved the β-amylolysis efficiency. Importantly, ISO could simultaneously hydrolyze the inner and outer branching points of amylopectin, and its catalytic activity was not inhibited by maltose.

## 4. Conclusions

In this study, the hydrolysis activity of AMY to different substrates showed obvious differences, and the most suitable substrate was maltodextrin. AMY could hydrolyze most of the α-1,6-glycosidic bonds in maltodextrin and hydrolyze the polymerized amylopectin to form small molecular linear glucans and oligosaccharides, but could not hydrolyze α-1,4-glycosidic bonds. AMY had a strong substrate selectivity, which tends to hydrolyze medium-length branched structures. The synergistic effect of AMY and β-amylase on DE6 maltodextrin could increase the hydrolysis rate to 86%, but there was no significant difference between the hydrolysis rate of waxy corn starch and cassava starch. The optimal debranching length of AMY was DP 13–24, which is more concentrated than that of PUL and shorter than that of ISO. In the production of different starch sugars, it could be necessary to add debranching enzymes with different substrate specificities to significantly improve the saccharification efficiency. However, the substrate of the saccharification is not the optimal substrate for ISO, and ISO cannot achieve its maximum debranching efficiency; the glucose produced by PUL debranching could inhibit the conversion of non-glucose starch sugars. The optimal debranching length of AMY was consistent with the branching length of the glycated substrate. In the production of oligosaccharides, the synergistic effect of AMY and glucoamylase could effectively relieve the blocking effect of branch points on α-1,4-glycoside hydrolase, thereby improving the yield of oligosaccharides.

## Figures and Tables

**Figure 1 foods-11-01442-f001:**
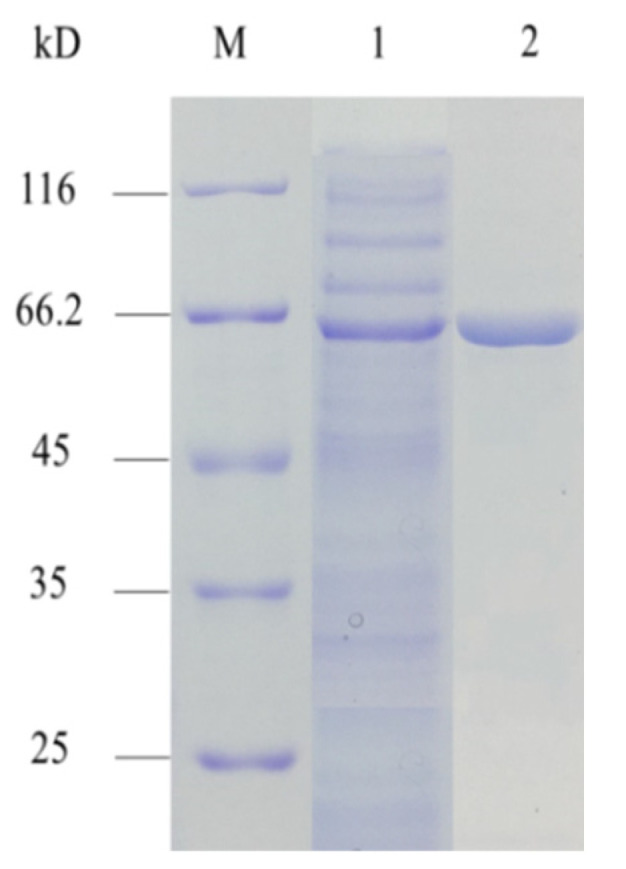
SDS-PAGE analysis of purified AMY. M: Molecular weight marker; 1: Crude enzyme; 2: Elution by 30% buffer B.

**Figure 2 foods-11-01442-f002:**
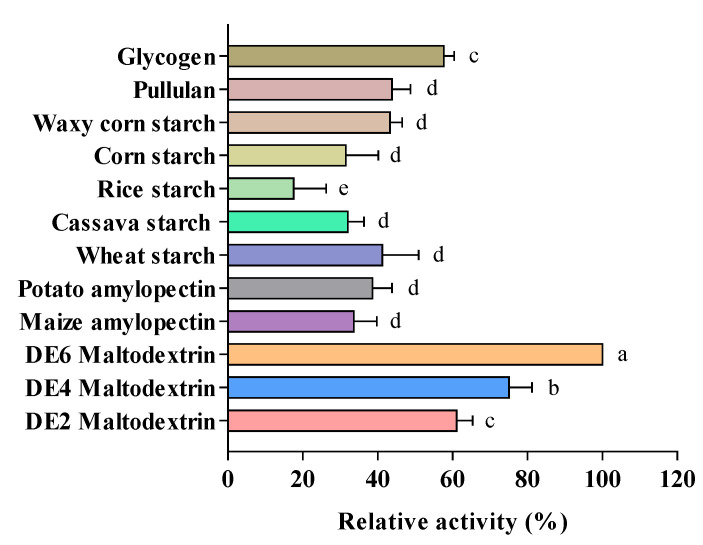
Substrate specificity of purified AMY. All data are means ± SD (*n* = 3). Means with different letters within the same group are significantly different (*p* < 0.05).

**Figure 3 foods-11-01442-f003:**
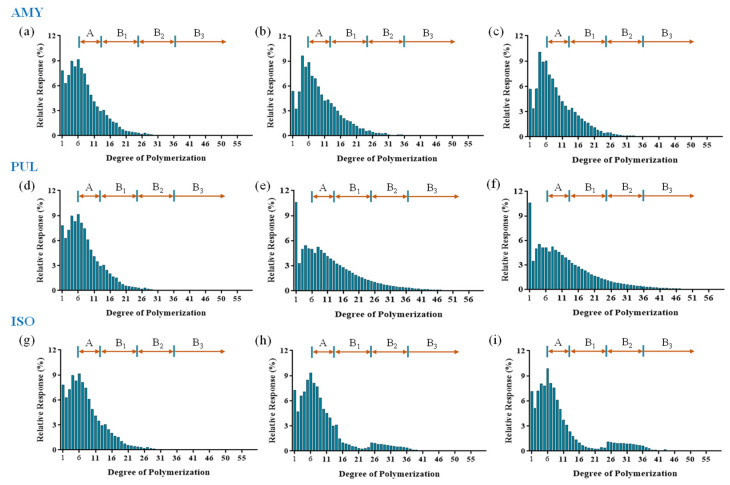
Chain length distribution of hydrolysates was obtained by debranching DE6 maltodextrin using AMY, PUL, and ISO, respectively. (**a**–**c**) were treated by AMY with 0, 6 and 24 h; (**d**–**f**) were treated by PUL with 0, 6 and 24 h; (**g**–**i**) were treated by ISO with 0, 6 and 24 h.

**Figure 4 foods-11-01442-f004:**
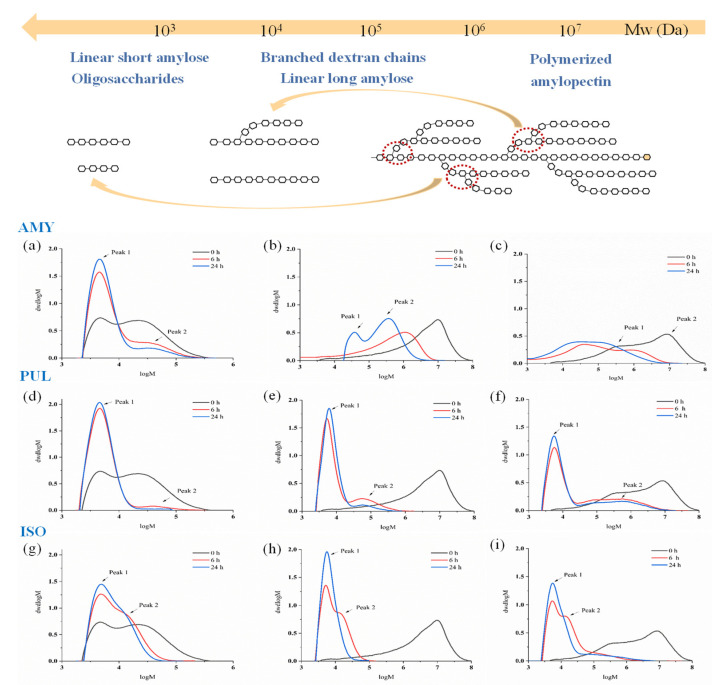
The molecular weight distribution of hydrolysates was obtained by debranching DE6 maltodextrin, waxy corn starch, and cassava starch using AMY, PUL, and ISO, respectively. The substrates of (**a**,**d**,**g**) were DE6 maltodextrin, (**b**,**e**,**h**) were waxy corn starch, (**c**,**f**,**i**) were cassava starch.

**Figure 5 foods-11-01442-f005:**
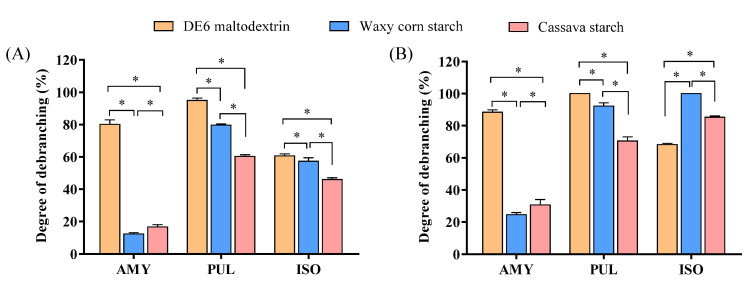
Debranching degree of dextrin and starch by AMY, PULand ISO under different treatment times. (**A**) debranched for 6 h; (**B**) debranched for 24 h. Means with the asterisk within the same group are significantly different (*p* < 0.05).

**Figure 6 foods-11-01442-f006:**
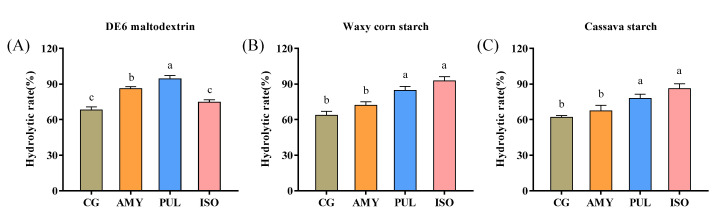
Degree of β-amylolysis limit of hydrolysates obtained by debranching DE6 maltodextrin, waxy corn starch, and cassava starch using AMY, PUL, and ISO, respectively. The substrate of (**A**) was DE6 maltodextrin, (**B**) was waxy corn starch, and (**C**) was cassava starch. Means with different letters within the same group are significantly different (*p* < 0.05).

**Table 1 foods-11-01442-t001:** Purification of AMY.

Component	Total Enzyme Activity (U)	Total Protein Content (mg)	Specific Enzyme Activity (U/mg)	Recovery Rate (%)
Crude enzyme	9853.0	42.0	234.6	-
HisTrap	3666.5	5.0	733.3	37.2

**Table 2 foods-11-01442-t002:** Chain length distribution of hydrolysates was obtained by debranching DE6 maltodextrin with AMY, PUL, and ISO, respectively.

Sample	Time	Chain Length Distribution (%) ^1^					CL ^3^
		DP ^2^ < 6	DP 6–12	DP 13–24	DP 25–36	DP ≥ 37	
AMY	0 h	38.57 ± 0.27 ^a^	43.20 ± 0.15 ^b^	17.09 ± 0.34 ^d^	1.13 ± 0.12 ^e^	0.05 ± 0.01 ^d^	7.97 ± 0.27 ^c^
	6 h	31.74 ± 0.65 ^d^	42.20 ± 0.48 ^c^	23.12 ± 0.22 ^b^	2.76 ± 0.17 ^e^	0.19 ± 0.02 ^d^	9.27 ± 0.41 ^b^
	24 h	33.68 ± 0.37 ^c^	41.89 ± 0.61 ^c^	21.88 ± 0.27 ^c^	2.39 ± 0.19 ^d^	0.14 ± 0.01 ^d^	9.37 ± 0.39 ^b^
PUL	0 h	38.57 ± 0.27 ^a^	43.20 ± 0.15 ^b^	17.09 ± 0.34 ^d^	1.13 ± 0.12 ^d^	0.05 ± 0.01 ^d^	7.97 ± 0.27 ^c^
	6 h	29.36 ± 0.31 ^e^	32.16 ± 0.28 ^d^	27.53 ± 0.16 ^a^	8.22 ± 0.47 ^b^	2.76 ± 0.38 ^a^	12.02 ± 0.62 ^a^
	24 h	28.90 ± 0.42 ^e^	32.41 ± 0.37 ^d^	27.53 ± 0.39 ^a^	8.30 ± 0.27 ^b^	2.88 ± 0.19 ^a^	12.11 ± 0.50 ^a^
ISO	0 h	38.57 ± 0.27 ^a^	43.20 ± 0.15 ^b^	17.09 ± 0.34 ^d^	1.13 ± 0.12 ^e^	0.05 ± 0.01 ^d^	7.97 ± 0.27 ^c^
	6 h	34.10 ± 0.67 ^c^	44.94 ± 0.47 ^a^	12.45 ± 0.37 ^e^	7.76 ± 0.19 ^c^	0.75 ± 0.01 ^c^	9.33 ± 0.47 ^b^
	24 h	35.29 ± 0.52 ^b^	43.45 ± 0.38 ^b^	9.61 ± 0.21 f	10.27 ± 0.23 ^a^	1.38 ± 0.02 ^b^	9.86 ± 0.34 ^b^

^1^ Distributions were calculated as the relative peak area (%); ^2^ DP, degree of polymerization; ^3^ CL, average chain length. All data are means ± SD (*n* = 3). Means with different letters within the same column are significantly different (*p* < 0.05).

**Table 3 foods-11-01442-t003:** Molecular weight distributions of hydrolysates were obtained by debranching DE6 maltodextrin, waxy corn starch, and cassava starch with AMY, PUL, and ISO, respectively.

Sample	Time	DE6 Maltodextrin		Waxy Corn Starch		Cassava Starch	
		Peak1 *M*_W_ ^1^ (×10^3^)	Peak2 *M*_W_ (×10^4^)	Peak1 *M*_W_ (×10^5^)	Peak2 *M*_W_ (×10^6^)	Peak1 *M*_W_ (×10^5^)	Peak2 *M*_W_ (×10^6^)
AMY	0 h	6.21 ± 0.13	5.30 ± 0.21	n.d. ^2^	7.44 ± 0.67	3.40 ± 0.57	9.84 ± 0.76
	6 h	5.81 ± 0.16	5.24 ± 0.19	0.57 ± 0.09	2.16 ± 0.11	0.58 ± 0.86	1.28 ± 0.22
	24 h	5.31 ± 0.11	4.26 ± 0.15	0.36 ± 0.13	0.94 ± 0.34	0.39 ± 0.23	0.94 ± 0.34
PUL	0 h	6.21 ± 0.13	5.30 ± 0.21	n.d.	7.44 ± 0.67	3.40 ± 0.57	9.84 ± 0.76
	6 h	5.48 ± 0.23	5.31 ± 0.18	0.07 ± 0.05	0.11 ± 0.08	0.07 ± 0.02	0.90 ± 0.42
	24 h	5.51 ± 0.27	n.d.	0.07 ± 0.03	0.08 ± 0.02	0.07 ± 0.01	0.87 ± 0.22
ISO	0 h	6.21 ± 0.13	5.30 ± 0.21	n.d.	7.44 ± 0.67	3.40 ± 0.57	9.84 ± 0.76
	6 h	5.52 ± 0.17	1.97 ± 0.24	0.01 ± 0.00	0.02 ± 0.01	0.09 ± 0.07	0.42 ± 0.15
	24 h	5.53 ± 0.22	1.66 ± 0.35	0.01 ± 0.00	n.d.	0.06 ± 0.22	0.08 ± 0.05

^1^*M*_W_: weight-average molar mass; ^2^ n.d., not detectable. All data are means ± SD (*n* = 3).

## Data Availability

The data presented in this study are available on request from the corresponding author.

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
