# Peer review of "Substrate Selectivity of a Novel Amylo-α-1,6-glucosidase from Thermococcus gammatolerans STB12"

_foods, 2022, doi:10.3390/foods11101442_

Round 1

Reviewer 1 Report

In my opinion, the manuscript is interesting, well-written, and scientifically sound, but some revision is necessary.

- The Authors should display the main goal of the present study well in the introduction section

- Some results and discussion parts  should be provided with additional references

- Manuscript has some grammatical errors, please check.

Reviewer 2 Report

  This article deals with the evaluation of Amylo-α-1,6-glucosidase enzyme property from Thermococcus gammatolerans STB12. Especially, the comparison of substrate selectivity and debranching efficiency were also analyzed in different kind of enzyme. These results present high economic value for its application. But there are several issues which needs to be addressed.

  1. The data of fig. 2 and Table 1 are the same. There is no need to give number for Chain length distribution. The figure can give more information for its distribution trend. Therefore, I suggested that the table 1 can be removed. For the same reason, the table 2 can be removed, too.
  2. The detail information of all substrates such as molecular weight should be revealed in the materials and methods section.
  3. The information of AMY such as its source, purification process, purity and MW should also be added.
  4. The microorganism “Thermococcus gammatolerans” was mentioned in the title of the paper, but no any background or information were provided in the manuscript.
  5. Substrate specificity is the main evaluation in this study. Please explain why there is no substrate specificity analysis of PUL and ISO in fig. 1?

Reviewer 3 Report

General remarks to the manuscript:

  • The amylo-α-1,6-glucosidase (AMY) is the one of basic enzymes used for debranching polysaccharides. In the proposed study Authors research on novel amylo-α-1,6-glucosidase from Thermococcus gammatolerans However, in this case, some essential details are missing. First of all, Authors are asked to implement the information about this source of AMY in introduction. How is this more advantageous than using other microbial strains described so far in the literature? The purpose of the study should be more emphasized. There should be mentioned that the research concerns working with a novel enzyme.
  • The procedure of isolation of the AMY from Thermococcus gammatolerans STB1 is missing in the methodology section and this description should be inserted.
  • Authors are asked to explain why they haven’t compare the substrate specificity of AMY from Thermococcus gammatolerans STB12 with amylo-α-1,6-glucosidases from other sources? In my opinion, this would be more reasonable and could reveal differences in the substrate specificity of enzymes with the same catalytic properties (cleavage of α-1,6-glycosidic bonds) and possibly indicate the benefits of this novel enzyme. What was the main motivation that Authors used pullulanase and isoamylase for comparison purpose?
  • Conclusion section should be improved. In particular, Authors are asked to insert the information how the varied substrate specificity of the debranching enzymes (including AMY) could be used in industrial processes.
  • Authors are asked to update the reference list, particularly with the articles from 2022. In current form of the manuscript there is no citation from 2022 at all.

Detailed remarks to the manuscript:

  • Line 66 – change ‘AMY had …’ to ‘AMY has…’
  • Line 68 – after the phrase ‘…explore the expression and physicochemical properties’ insert ‘of this enzyme.’
  • Line 81 – insert ‘novel’ before ‘AMY’
  • Subsection 2.1 – some reagents are missing here, please complete.
  • Subsection 2.3 – the calibration curve for glucose concentration determination is missing here. Authors are asked to insert at least the equation of the standard curve.
  • Authors are asked to explain why the procedures of enzyme activity determination are different in point 2.2 and 2.3.
  • Line 171 – put the formulas used for calculations
  • Line 180 – the word ‘polysaccharide’ itself is too general; put the detailed name of the substrate or list all the substrates used in the research here.
  • Table 1 and Table 2 – insert the a lines between rows for individual enzymes
